# The Role of Veracity on the Load Monitoring of Professional Soccer Players: A Systematic Review in the Face of the Big Data Era

**João Gustavo Claudino** [1,2,*], **Carlos Alberto Cardoso Filho** [1], **Daniel Boullosa** [3,4,5], **Adriano Lima-Alves** [6], **Gustavo Rejano Carrion** [1], **Rodrigo Luiz da Silva GianonI** [2], **Rodrigo dos Santos Guimarães** [7,8], **Fúlvio Martins Ventura** [9], **André Luiz Costa Araujo** [10], **Sebastián Del Rosso** [11], **José Afonso** [12] and **Julio Cerca Serrão** [1]

1   Laboratory of Biomechanics, School of Physical Education and Sport, Campus São Paulo, Universidade de São Paulo, São Paulo 05508-030, State of São Paulo, Brazil; carlos.filho@alumni.usp.br (C.A.C.F.); gustavo.carrion@usp.br (G.R.C.); jcserrao@usp.br (J.C.S.)

2   Research and Development Department, LOAD CONTROL, Contagem 32280-440, Minas Gerais, Brazil; rodrigogianoni@yahoo.com.br

3   College of Healthcare Sciences, Campus Townsville, James Cook University, Townsville 4811, Australia; daniel.boullosa@gmail.com

4   INISA, Graduate Program of Movement Sciences, Campus Campo Grande, Federal University of Mato Grosso do Sul, Campo Grande 79070-900, Mato Grosso do Sul, Brazil

5   Research and Development Department, iLOAD Solutions, Campo Grande 79118-280, Mato Grosso do Sul, Brazil

6   Department of Physiology, Fortaleza Esporte Clube, Fortaleza 60510-290, Ceará, Brazil; adrianolimaa@gmail.com

7   Department of Physiology, América Futebol Clube, Contagem 32180-060, Minas Gerais, Brazil; rodrigo.sportcoach@gmail.com

8   Program de Doctorado en Ciencias del Deporte, Facultad de Ciencias del Deporte, Universidad de Extremadura, 10071 Cáceres, Spain

9   Department of Performance, Guarani Futebol Clube, Campinas 13100-200, State of São Paulo, Brazil; fulvio.ventura@gmail.com

10  Health and Performance Department, GoiásEsporte Clube, Goiânia 74343-010, Goiás, Brazil; andrearaujo118@hotmail.com

11  Centro de Investigaciones en Nutrición Humana, Escuela de Nutrición, Facultad de CienciasMédicas, Campus Córdoba, Universidad Nacional de Córdoba, Córdoba 5000, Argentina; delrossosebastian@gmail.com

12  Centre for Research, Education, Innovation and Intervention in Sport, Faculty of Sport, Campus Porto, University of Porto, 4099-002 Porto, Portugal; jneves@fade.up.pt

*   Correspondence: claudinojgo@usp.br

**Abstract:** Big Data has real value when the veracity of the collected data has been previously identified. However, data veracity for load monitoring in professional soccer players has not been analyzed yet. This systematic review aims to evaluate the current evidence from the scientific literature related to data veracity for load monitoring in professional soccer. Systematic searches through the PubMed, Scopus, and Web of Science databases were conducted for reports onthe data veracity of diverse load monitoring tools and the associated parameters used in professional soccer. Ninety-four studies were finally included in the review, with 39 different tools used and 578 associated parameters identified. The pooled sample consisted of 2066 footballers (95% male: $24 \pm 3$ years and 5% female: $24 \pm 1$ years). Seventy-three percent of these studies did not report veracity metrics for anyof the parameters from these tools. Thus, data veracity was found for 54% of tools and 23% of parameters. The current information will assist in the selection of the most appropriate tools and parameters to be used for load monitoring with traditional and Big Data approaches while identifying those still requiring the analysis of their veracity metrics or their improvement to acceptable veracity levels.

**Keywords:** data analytics; fitness; illness; injury; performance; recovery

## 1. Introduction

Load monitoring is used to ensure the best workload that optimizes physical fitness and sports performance while preventing athletes from injury or illness in soccer [1]. In this context, a true individualization of all training factors in team sports is mandatory for a better fitness–fatigue equilibrium through an individualized monitoring approach [2]. Meanwhile, the number of injuries [3] and the physical performance of soccer players continue to increase over the years [4]. For instance, in a 13-year longitudinal analysis, it was found that hamstring injuries have annually increased by 4% in professional soccer players [3]. Factors associated with load monitoring, such as player load, match frequency, playing style, team management, and the continuity of technical staff, have also influenced these trends [3]. On the other hand, it is well accepted that physical performance in soccer matches has continuously improved over the past years. For instance, total distance covered, high-intensity running and sprinting distances, and number of sprints increased by 2% (effect size (ES): 0.22), 30% (ES: 0.82), 35% (ES: 0.93), and 85% (ES: 1.46), respectively, across 7 seasons of the English Premier League (i.e., 2006–2007 compared to 2012–2013) [4]. However, the $VO_{2max}$ of elite female [5] and male [6] soccer players has not changed between 1989 and 2007 for females and between 1989 and 2012 for males. This paradox of a similar aerobic fitness between decades, despite objectively increased match demands, highlights the need to select appropriate monitoring tools. Furthermore, considering that soccer is a complex sport in which players need to develop several physical capacities (e.g., acceleration, agility, endurance), the selection of the best tools to monitor the evolution of players' physical fitness over the season is required for better managing the complex balance between training, competition, recovery, and evaluation [7–10].

We are living in a time where technology has experienced a fast evolution in the sports arena [11]. This technological growth has enabled the 24 h monitoring of athletes on an individual basis [12], thus allowing large data sets, i.e., Big Data, to be gathered in sport settings. Careful analyses of these large datasets can enhance our knowledge in sportsscience and medicine, thus supporting the making-decision process for designing appropriate training and competitive strategies [13,14]. However, some premises, which are known as the four Vs of Big Data, should be attained fromthese datasets for these purposes: (1) volume, (2) variety, (3) velocity, and (4) veracity [15]. Volume refers to the size of the datasets; variety refers to different data formats and data sources; velocity describes the speed at which data is generated and processed for analyses; and veracity refers to the accuracy, quality, relevance, uncertainty, reliability, and predictive values of the collected data [14–17]. Focusing on veracity to identify the best monitoring tools and parameters in soccer is mandatory to identify the "errors" [18]. Furthermore, veracity is a preliminary step to effectively meet all these premises and, thus, optimize the applicability of Big Data to soccer with the best cost-to-benefit ratio. For instance, according to IBM [15], poor data quality generates an economic cost of around USD 3.1 trillion per year in the USA. If this value were the gross domestic product (GDP) of any country, it would be placed in the top 6 GDPs in the world.

Therefore, this systematic review aims to evaluate data veracity for load monitoring in professional soccer. This information is important for a deep understanding of the tools and parameters used in load monitoring as well as the effective implementation of Big Data analyses in professional clubs.

## 2. Materials and Methods

We adopted the Preferred Reporting Items for Systematic Reviews and Meta-Analyses (PRISMA) guidelines [19].

### 2.1. Sources and Study Selection Process

Three electronic databases (PubMed, Scopus, and Web of Science) were systematically searched from inception to August 2020. The command line ("monitoring" OR "monitor") AND ("training load" OR "load") AND ("soccer" OR "football") was used in the

title, abstract, and keywords during the electronic searches. The selection process was completed by two authors (J.G.C. and C.A.C.F.). If some doubt was found during this step, a third author (D.B.) assisted in the decision. The quality appraisal was completed by one author (J.G.C.).

### 2.2. Eligibility Criteria

Initially, the articles reviewed were identified after reading the titles and abstracts based on the following inclusion criteria:

(1) The study was written in English;
(2) The study was published as original research in a peer-reviewed journal, and a full-text article was available;
(3) Data were reported for soccer players;
(4) The participants were professional soccer players;
(5) Load monitoring parameters were included.

Then, after this first screening, the eligibility criteria according to PECO were applied in the remaining full manuscripts. (P)articipants: healthy, professional soccer players of any age and sex. (E)xposure: exposure to load monitoring in the training session and/or on match day. (C)omparators: control groups without load monitoring are accepted but not mandatory. (O)utcomes: load monitoring tools and parameters.

### 2.3. Data Extraction

Five authors (A.L.M., G.R.C., R.G., F.M.V., and S.D.R.) extracted the following information from the included full-text articles: authors, year, sample information (mean, age, sex, sample size, competitive level), study design, veracity data, and information on load monitoring tools and parameters. The parameters were classified according to the applied method (e.g., laboratory or portable methods; venous or capillary blood samples). Discrepancies were resolved through discussion until consensus was reached. A narrative synthesis of data was performed.

The median was used when a range of a veracity metric was reported in a manuscript. All veracity metrics of the reported parameters were used to determine the median and range (minimum–maximum) for each tool.

### 2.4. Quality Assessment

The quality of all studies was evaluated by one author (J.G.C.) using objective evaluation criteria (see supplementary material, Table S1) based on a previous study by Saw et al. [20]. Scores were assigned based on how well each criterion was met, assuming a maximum possible score of 8 (low risk of bias). Studies with a score of $\leq 4$ were considered poor and were, therefore, subsequently excluded.

### 2.5. Veracity Analysis

Veracity analysis was performed based on the use of measures related to accuracy, uncertainty, reliability, and quality of data [18,21–23]. All these measures were only obtained when explicitly reported by the authors' own data. For the used instruments, accuracy was checked in the manufacturer's official documentation or website. Accuracy is understood as the closeness of the agreement between the result of a measurement and the true value of the measurand [24]. Uncertainty and reliability were obtained by absolute and relative consistencies using the standard error of measurement (SEM) [25] (also known as the typical error of measurement (TEM) or typical error (TE) [26], beyond the measures generated from the SEM itself as the minimum difference (MD) or minimum detectable change (MDC)) and the intraclass correlation coefficient (ICC) [25–27]. The SEM is the measure of the amount of error variance in a set of obtained scores, different from the standard error of estimate (SEE), which is the standard deviation of true scores if the observed score is held constant [28]. Standard uncertainty is the uncertainty of the result of a measurement expressed as a standard deviation [29]. Precision is understood as the closeness of agreement between the

results of successive measurements of the same measurand, carried out under the same conditions of measurement. Precision is also called repeatability [24]. Limits of agreement (LOAs) refer to the reference interval for the test–retest differences expected for 95% of the population [30]. The coefficient of variation (CV) refers to the variation around the average, expressed as a percentage; data quality by means of the CV is interpreted by the level of instability [31].

Some reference values are assumed to better understand the results. SEM results lower than 5% have been suggested [32,33] and classified as good (SEM = < 5.0%), moderate (SEM = 5.0–9.9%), or poor (SEM $\geq$ 10.0%) [34–36]. However, it is recommended to present the corresponding value with confidence intervals (CIs) [37,38] because this is a measure of how much the measured test scores are spread around a "true" score [37,38]. Although the SEM may be better than the ICC [25] to evaluate reliability, both are reported in the present manuscript. According to Koo and Li [39], ICC values < 0.50 = poor, between 0.50 and 0.75 = moderate, between 0.75 and 0.90 = good, and >0.90 =indicative of excellent reliability. For the CV, it has been suggested that CV > 30.0% = large, CV < 30.0% and >10.0% = medium, and CV < 10.0% = small [40]. These reference values were used in the figure to determine the traffic-light systems(i.e., red, amber or green). For accuracy, MDC, SEE, precision, uncertainty and LOAs, no reference values were found in the scientific literature.

## 3. Results

The initial search returned 1035 articles (see Figure 1). After the removal of duplicates (n = 461), a total of 574 studies were retained for full-text screening. Following the eligibility assessment, 479 studies were excluded as they did not meet the inclusion criteria, while one full-text manuscript was not found. Finally, 94 studies were included in this systematic review [41–134].

### 3.1. Characteristics of the Studies and Risk of Bias

The pooled sample size and age included 2066 participants aged 24 $\pm$ 3 years, being composed mostly of male athletes (95%). The average duration of the load monitoring interventions was 168 days (range: 1–1034). The athletes were all professional soccer players from first divisions (81%), second divisions (12%), or third divisions (1%), andnational teams (6%). A single study (1%) did not report the athletes' level. Regarding the nationality of the soccer players, they were from England (16%), Spain (15%), Brazil (9%), Australia (7%), Portugal (7%), Netherlands (6%) Italy (6%), France (4%), Denmark (2%), Qatar (2%), Tunisia (2%), Russia (2%), Iran (2%), Austria (1%), Belgium (1%), Chile (1%), Colombia (1%), Czech Republic (1%), Germany (1%), Latvia (1%), Norway (1%), Singapore (1%), Switzerland (1%), and Wales (1%). Seven studies did not report the nationality of the athletes but reported the continent (Europe = 7%).

The average bias score for the studies was 6.6 (range: 4–8). Only one study had a high risk of bias, with a score $\leq$4 (see Table S1). There were 88 observational studies (i.e., 94%), with 82 prospective cohort studies [41,43–49,51,52,54–56,59–61,63–84,86–88,90,92–109,111–114,116–123,125–134], 3 retrospective cohort studies [85,110,115], 2 cross-sectional studies [50,62], and one diagnostic accuracy study [124]. There were six experimental studies (i.e., 6%), with three pre–post interventional studies [42,58,89], two cross-over randomized controlled trials [53,91], and one non-randomized trial [57] (see Table S2).

### 3.2. Main Findings

Most studies, i.e., 73%, did not report any veracity metric for all the identified parameters (n = 69). At least one veracity metric was reported in 18% of studies (n = 17), with a veracity metric for all parameters reported in 9% of studies (n = 8; see Figure **??**). Thirty-nine different tools were used for load monitoring: electronic performance tracking systems (EPTS) = 64% (n = 60), rating of perceived exertion (RPE) = 48% (n = 45), heart rate monitoring (HR) = 29% (n = 27), countermovement jump test (CMJ) = 12% (n = 11), psycho-

metric questionnaire with Likert scale = 12% (n = 11), blood samples = 6% (n = 6), maximal running test = 6% (n = 6), salivary imunoglobulina A (s-IgA) = 6% (n = 6), blood lactate concentration = 5% (n = 5), video-computerized systems = 5% (n = 5), watch = 5% (n = 5), body composition = 3% (n = 3), integrative tool of training load assessment = 3% (n = 3), repeated sprint ability = 3% (n = 3), salivary cortisol = 3% (n = 3), salivary testosterone = 3% (n = 3), submaximal running test = 3% (n = 3), sprint test = 3% (n = 3), anaerobic speed reserve = 2% (n = 2), creatine kinase (CK) = 2% (n = 2), maximal oxygen uptake = 2% (n = 2), Visual Analog Scale Questionnaire = 2% (n = 2), total quality recovery (TQR) = 2% (n = 2), training planning = 2% (n = 2), actigraphy = 1% (n = 1), blood ammonia concentration = 1% (n = 1), coach rating of performance = 1% (n = 1), Edinburgh Mental Well-being Scale = 1% (n = 1), infrared thermography = 1% (n = 1), isometric force testing = 1% (n = 1), Neuro-muscular Efficiency Index = 1% (n = 1), oxidative stress = 1% (n = 1), Physical Activity Enjoyment Scale = 1% (n = 1), Profile of Mood State Questionnaire (POMS) = 1% (n = 1), RESTQ-Sport Scale = 1% (n = 1), sit-and-reach test = 1% (n = 1), squat jump test (SJ) = 1% (n = 1), tensiomyography = 1% (n = 1), and urine metabolomic = 1% (n = 1).

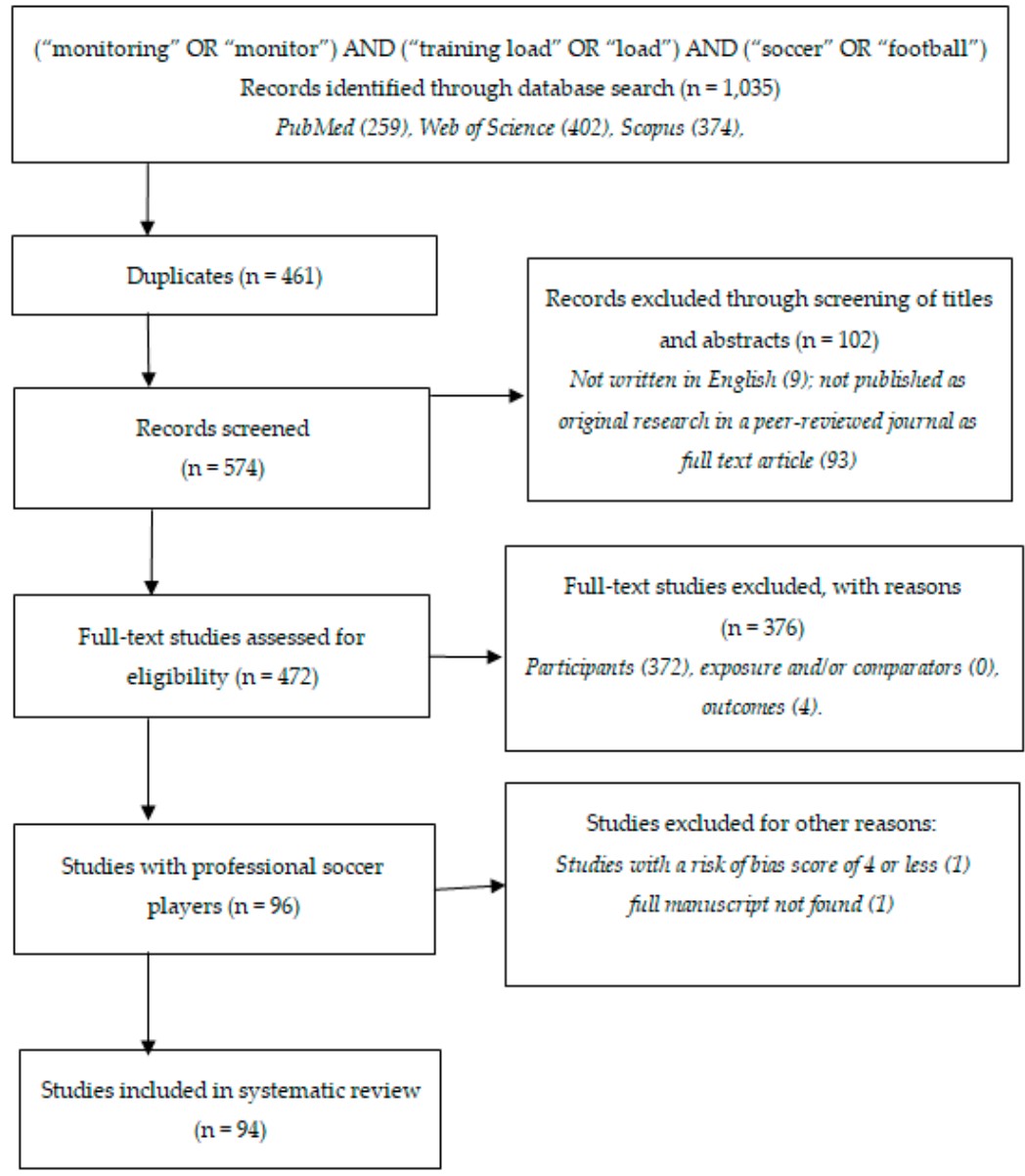

**Figure 1.** Study selectionin a PRISMA flow diagram.

A total of 578 parameters were used to monitor the load from these 39 tools. Data veracity was reported for 54% of the tools in at least one parameter, which resulted in 23% of these parameters. Thus, most studies did not report metrics of veracity for their tools and parameters (see details in Figure **??**, Table 1 and Table S2). Specifically, Table S2 presents a summary of all the selected studies, with further information on study design (and duration), sample level (n; sex; age; country of sample), tools (brand and model or reference), accuracy reported by the company, and parameters as well as veracity analyses (metrics).

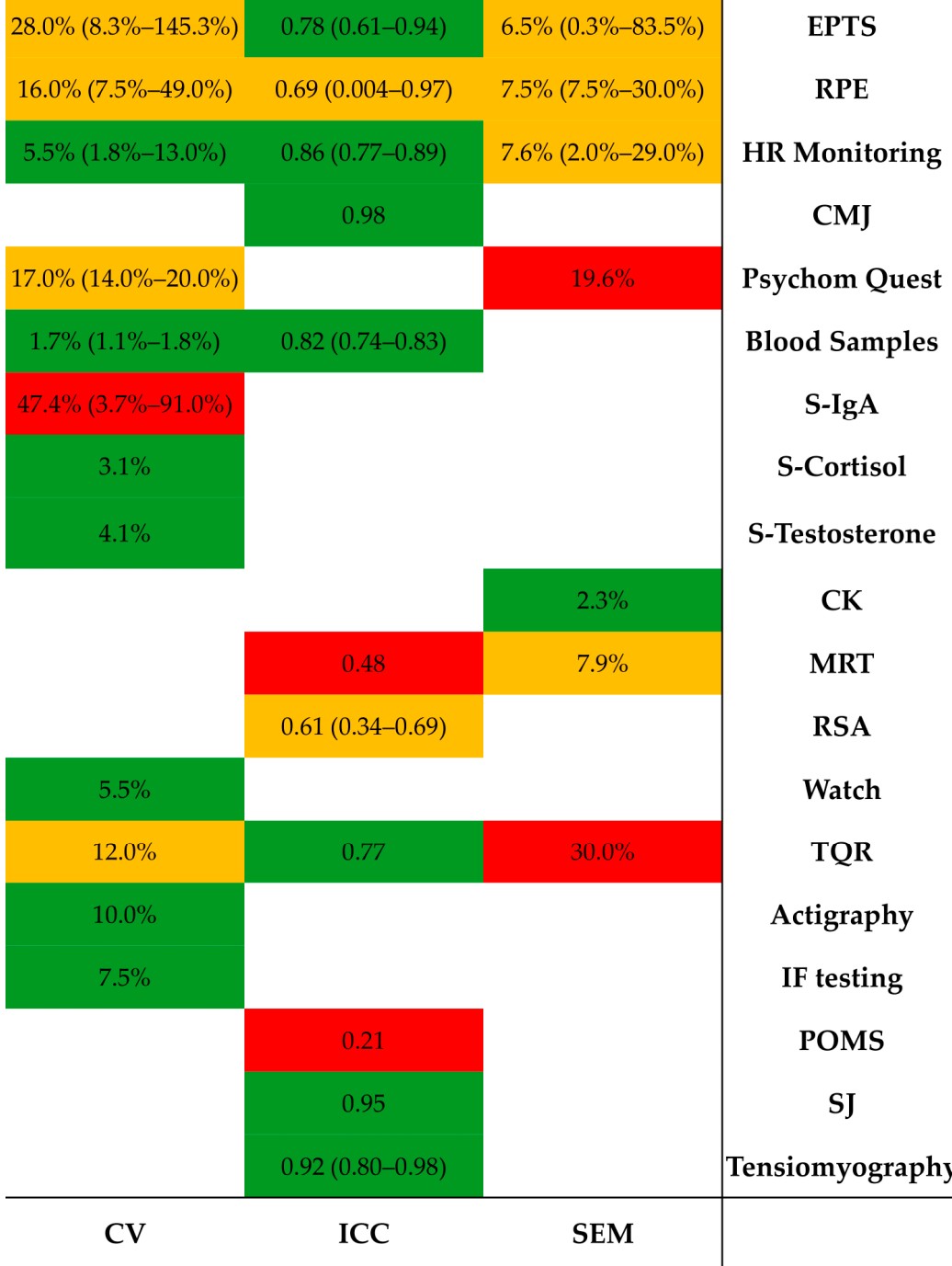

**Figure 2.** Veracity metrics classifications.Median (minimum–maximum) and result (i.e., median) without range (i.e., minimum and maximum) because it was found just for one value (n = 1). Based on the current method, the respective reference values were used in this figure to determine the traffic-light systems (i.e., red, amber, or green). The median was used to

determine the color, and blank cells are due to the lack of veracity metrics for that case. CV = coefficient of variation (standard deviation divided by the mean, expressed as a percentage, %); ICC = intraclass correlationcoefficient; SEM = standard error of measurement; EPTS = electronic performance tracking system; RPE = rating of perceived exertion; HR = heart rate; CMJ = countermovement jump; Psychom Quest = Psychometric Questionnaire with Likert scale; S-IgA = salivary imunoglobulina A; S-Cortisol = salivary cortisol; S-Testosterone = salivary testosterone; CK = creatine kinase; MRT = maximal running test; RSA = repeated sprint ability; TQR = total quality recovery; IF = isometric force; POMS = Profile of Mood State Questionnaire; SJ = squat jump; Watch = wristwatch.

**Table 1.** Veracity metrics ranking.

| Tools (n = 39): 54% (21/39) with Veracity Metrics | Parameters (n = 578): 23% (131/578) with Veracity Metrics | | Veracity Metrics (Classification) | |
|---|---|---|---|---|
| | Total distance covered (m) | (n = 8) | SEM = 0.3–5.3% CV = 8.3–43.0% ICC = 0.94 Bias = 0.8 ± 5.4 and 2–6% SEE = 4–5% | (Good–Moderate) (Small–Large) (Excellent) |
| | Average speed (m/min) | (n = 3) | SEM = 4.2% CV = 16.0–19.0% | (Good) (Medium) |
| | PlayerLoad (AU) | (n = 3) | SEM = 0.6% CV = 20.0–37.0% | (Good) (Medium–Large) |
| | Top speed (km/h) | (n = 3) | SEM = 2.6–7.7% CV = 10% | (Good–Moderate) (Small) |
| | Distance covered at>21 km/h (m) | (n = 2) | SEM = 42.9–53.0% MDC = 48.0–168.0% | (Poor) |
| | High metabolic power distance (≥20 W/kg; m) | (n = 2) | SEM = 6.1% CV = 30.0% ICC = 0.78 Bias = 0.8 ± 1.9 | (Moderate) (Large) (Good) |
| | High-speed running (19.8–25.2 km/h; m) | (n = 2) | SEM = 6.8% Bias = 2–10% SEE = 10–11% | (Moderate) |
| | Maximum acceleration (>3 m/s$^2$; m) | (n = 2) | CV = 34.0–43.3% | (Large) |
| | Maximum deceleration (>−3 m/s$^2$; m) | (n = 2) | CV = 32.0–36.6% | (Large) |
| | PlayerLoad/min (AU/min) | (n = 2) | SEM = 4.5% CV = 20.0% | (Good) (Medium) |
| | Sprinting distance covered (>25.2 km/h; m) | (n = 2) | SEM = 9.7% Bias = 4–10% SEE = 14–22% | (Moderate) |
| | Accelerations between 2.5–4.0 m/s$^2$ (number) | (n = 1) | SEM = 16.3–18.1% | (Poor) |
| | Accelerations >3 m/s$^2$ (number) | (n = 1) | CV = 68.0% | (Large) |
| | Accelerations ≥3.3 m/s$^2$ (number) | (n = 1) | SEM = 14.1% ICC = 0.61 Bias = 0.3 ± 2.5 | (Poor) (Moderate) |
| | Accelerations >4 m/s$^2$ (number) | (n = 1) | SEM = 76.8–89.7%, | (Poor) |
| | Average metabolic power (W/kg) | (n = 1) | SEM = 3.4% ICC = 0.82 Bias = 1.3 ± 7.5 | (Good) (Good) |
| 1 | **EPTS** in 64% (60/94) of Studies and 20% (59/294) of Parameters with Veracity Metrics | | | |
| | Deceleration effort (>−3 m/s$^2$; number) | (n = 1) | CV = 70.0% | (Large) |
| | Decelerations (≥−3.3 m/s$^2$; number) | (n = 1) | SEM = 16.2% ICC = 0.67 Bias = 0.1 ± 2.9 | (Poor) (Moderate) |
| | Distance at acceleration zone count 2–3 (2 m/s$^2$– 3 m/s$^2$; m) | (n = 1) | CV = 30.0% | (Large) |
| | Distance at deceleration zone count 2–3 (−2–−3 m/s$^2$; m) | (n = 1) | CV = 31.0% | (Large) |
| | Distance covered between 13–18 km/h (m) | (n = 1) | SEM = 13.3–17.4% | (Poor) |
| | Distance covered between 17–21 km/h (m) | (n = 1) | MDC = 141.0–146.0% | |
| | Distance covered between 18–21 km/h (m) | (n = 1) | SEM = 21.0–22.5% | (Poor) |
| | Distance in Power Zone 20–25 w/kg (km) | (n = 1) | CV = 32.0% | (Large) |
| | Distance in Speed Zone 2 (no descriptions; km) | (n = 1) | CV = 27.0% | (Medium) |
| | Duration (min) | (n = 1) | CV = 35.0% | (Large) |
| | Dynamic stress load (AU) | (n = 1) | SEM = 2.5% ICC = 0.94 Bias = 0.8 ± 5.4 | (Good) (Excellent) |
| | Efforts/sprints >21 km/h (number) | (n = 1) | SEM = 44.5–49.0% | (Poor) |
| | High-intensity events (5.4–9.0 km/h; m) | (n = 1) | CV = 19.0% | (Medium) |
| | High-intensity events (9.0–12.6 km/h; m) | (n = 1) | CV = 17.0% | (Medium) |
| | High-intensity events (12.6–14.4 km/h; m) | (n = 1) | CV = 16.0% | (Medium) |
| | High-intensity running distance/min (>15.1 km/h; m/min) | (n = 1) | SEM = 30.6% | (Poor) |
| | High-intensity running distance (16–22 km/h; m) | (n = 1) | CV = 25.2% | (Medium) |
| | High-speed distance (19.8–25.5 km/h; m) | (n = 1) | SEM = 8.1% ICC = 0.65 Bias = 0.1 ± 1.2 | (Moderate) (Moderate) |

**Table 1.** *Cont.*

| | | Parameter | n | Metrics | Rating |
|---|---|---|---|---|---|
| | | High-speed running (19.8–25.1 km/h; %) | (n = 1) | CV = 96.5%<br>Bias = 2.3 | (Large) |
| | | High-speed running (individualized; %) | (n = 1) | CV = 57.0% | (Large) |
| | | High-speed running distance (14.4–19.8 km/h; m) | (n = 1) | CV = 13.0% | (Medium) |
| | | High speed running time (17–21 km/h; min) | (n = 1) | MDC = 123.0–141.0% | |
| | | Low-speed activities (<14.4 km/h; %) | (n = 1) | CV = 7.8%<br>Bias = 70.5 | (Small) |
| | | Low-speed activities (individualized; %) | (n = 1) | CV = 96.9% | (Large) |
| | | Moderate-speed running (14.4–19.8 km/h; %) | (n = 1) | CV = 53.4%<br>Bias = 69.2 | (Large) |
| | | Moderate-speed running (individualized; %) | (n = 1) | CV = 24.3% | (Medium) |
| | | PlayerLoad2D (AU) | (n = 1) | CV = 20.0% | (Medium) |
| | | PlayerLoad2D/min (AU•2D/min) | (n = 1) | SEM = 4.6% | (Good) |
| | | PlayerLoad AP (%) | (n = 1) | SEM = 3.9% | (Good) |
| | | PlayerLoad ML (%) | (n = 1) | SEM = 2.4% | (Good) |
| | | PlayerLoadm/min (AU•m/min) | (n = 1) | SEM = 2.8% | (Good) |
| | | PlayerLoadSlow/min (AU•Slow/min) | (n = 1) | SEM = 8.9% | (Moderate) |
| | | PlayerLoad V (%) | (n = 1) | SEM = 2.1% | (Good) |
| | | Power score (w/kg) | (n = 1) | CV = 19.0% | (Medium) |
| | | Sprinting (≥25.2 km/h; %) | (n = 1) | CV = 193.5%<br>Bias = 2.4 | (Large) |
| | | Sprinting (individualized; %) | (n = 1) | CV = 97.1% | (Large) |
| | | Sprint distance (>25.5 km/h; m) | (n = 1) | SEM = 16.1%<br>ICC = 0.77<br>Bias = 0.1 ± 2.9 | (Poor)<br>(Good) |
| | | Sprint distance (no descriptions; km) | (n = 1) | CV = 44.0% | (Large) |
| | | Total acceleration distance (m) | (n = 1) | CV = 18.3% | (Medium) |
| | | Total deceleration distance (m) | (n = 1) | CV = 15.9% | (Medium) |
| | | Very high-speed distance (>19.8 km/h; m) | (n = 1) | CV = 13.0% | (Medium) |
| | | Very high speed distance (>22 km/h; m) | (n = 1) | CV = 29.0% | (Medium) |
| | | Very high speedrunning time (>21 km/h; min) | (n = 1) | MDC = 160.0–305.0% | |
| 2 | **RPE**<br>in 48% (45/94) of Studies and 31% (12/39) of Parameters with Veracity Metrics | Foster session-RPE (AU)<br>(Scale 1–10 by Foster et al., 2001) | (n = 4) | SEM = 5.5%<br>CV = 18.0–49.0%<br>ICC = 0.57–0.77<br>Bias = 0.1 ± 1.2 | (Moderate)<br>(Medium–Large)<br>(Moderate–Good) |
| | | Borg RPE scores (Scale 6–20 by Borg, 1982) | (n = 1) | CV = 5.1–9.9% | (Small) |
| | | Carrie RPE score (Scale 0–10 by Carrie, 2012) | (n = 1) | CV = 19.0–31.0% | (Medium–Large) |
| | | Carrie session-RPE (Scale 0–10 by Carrie, 2012) | (n = 1) | CV = 47.0% | (Large) |
| | | Monotony (Scale 1–10 by Foster et al., 2001) | (n = 1) | ICC = 0.10 | (Poor) |
| | | Morandi session-RPE<br>(Scale 0–10 by Morandi et al., 2020) | (n = 1) | SEM = 23%<br>CV = 11.0%<br>ICC = 0.63 | (Poor)<br>(Medium)<br>(Moderate) |
| | | Morandi RPE score<br>(Scale 0–10 by Morandi et al., 2020) | (n = 1) | SEM = 30%<br>CV = 10.0%<br>ICC = 0.74 | (Poor)<br>(Small)<br>(Moderate) |
| | | Muscular session-RPE<br>(Scale 1–10 by Foster et al., 2001) | (n = 1) | SEM = 7.5%<br>CV = 15.7%<br>ICC = 0.97 | (Moderate)<br>(Medium)<br>(Excellent) |
| | | Respiratory session-RPE<br>(Scale 1–10 by Foster et al., 2001) | (n = 1) | SEM = 7.5%<br>CV = 16.2%<br>ICC = 0.96 | (Moderate)<br>(Medium)<br>(Excellent) |
| | | Strain (Scale 1–10 by Foster et al., 2001) | (n = 1) | ICC = 0.004 | (Poor) |
| | | Sum of all muscular RPE scores<br>(Scale 1–10 by Foster et al., 2001) | (n = 1) | CV = 14.5% | (Medium) |
| | | Sum of all respiratory RPE scores<br>(Scale 1–10 by Foster et al., 2001) | (n = 1) | CV = 14.0% | (Medium) |
| 3 | **HR Monitoring**<br>in 29% (27/94) of Studies and 18% (7/39) of Parameters with Veracity Metrics | Mean percentage of maximum HR (%) | (n = 3) | SEM = 2.2–27.0%<br>CV = 1.3–5.0%<br>ICC = 0.87–0.89<br>Bias = 0.2 ± 6.8 | (Good–Poor)<br>(Small)<br>(Good) |
| | | HR mean (bpm) | (n = 2) | SEM = 3.0–27.0%<br>CV = 5.0%<br>ICC = 0.77–0.89<br>Bias = 1.5 ± 10.4 | (Good–Poor)<br>(Small)<br>(Good) |
| | | Ln RMSSD (ms) | (n = 2) | SEM = 7.6%<br>CV = 6.0% | (Moderate)<br>(Small) |
| | | Banister's TRIMP (Banister, 1991) | (n = 1) | SEM = 29.0%<br>CV = 12.0%<br>ICC = 0.85 | (Poor)<br>(Medium)<br>(Good) |
| | | Maximal HR (bpm) | (n = 1) | SEM = 2.0%<br>ICC = 0.79<br>Bias = 0.3 ± 4.9 | (Good)<br>(Good) |
| | | Time above 85% HR$_{max}$(%) | (n = 1) | SEM = 2.2%<br>ICC = 0.85<br>Bias = 0.6 ± 5.4 | (Good)<br>(Good) |
| | | TRIMP$_{mod}$ (Stagno et al., 2007) | (n = 1) | SEM = 29.0%<br>CV = 13.0%<br>ICC = 0.87 | (Poor)<br>(Medium)<br>(Good) |

**Table 1.** *Cont.*

| | | | | | |
|---|---|---|---|---|---|
| 4 | **CMJ Test**<br>in 12% (11/94) of Studies and 8% (1/12) of Parameters with Veracity Metrics | Jump height (cm) | (n = 1) | ICC = 0.98 | (Excellent) |
| 5 | **Psychometric Questionnaire with Likert scale**<br>in 12% (11/94) of Studies and 56% (5/9) of Parameters with Veracity Metrics | Hooper Index<br><br>Energy Levels<br>Lower body soreness<br>Readiness to Train<br>Sleep quality | (n = 2)<br><br>(n = 1)<br>(n = 1)<br>(n = 1)<br>(n = 1) | SEM = 19.6%<br>CV = 15.0%<br>CV = 14.0%<br>CV = 20.0%<br>CV = 17.0%<br>CV = 20.0% | (Poor)<br>(Medium)<br>(Medium)<br>(Medium)<br>(Medium)<br>(Medium) |
| 6 | **Blood Samples**<br>in 6% (6/94) of Studies and 13% (6/48) of Parameters with Veracity Metrics | Cortisol<br>Plasma CK<br>Plasma CRP<br>Plasma LDH<br>Testosterone<br>T/C ratio | (n = 1)<br>(n = 1)<br>(n = 1)<br>(n = 1)<br>(n = 1)<br>(n = 1) | ICC = 0.74<br>CV = 1.8%<br>CV = 1.7%<br>CV = 1.1%<br>ICC = 0.83<br>ICC = 0.82 | (Moderate)<br>(Small)<br>(Small)<br>(Small)<br>(Good)<br>(Good) |
| 7 | **Maximal Running Test**<br>in 6% (6/94) of Studies and 40% (2/5) of Parameters with Veracity Metrics | Final velocity of the 30–15 Intermittent Fitness Test (Buchheit, 2008)<br>Total distance covered (m) Yo-Yo Intermittent Recovery Test Level 1 (Bangsbo et al., 2008) | (n = 1)<br><br>(n = 1) | SEM = 7.7–8.0%<br><br>ICC = 0.48 | (Moderate)<br><br>(Poor) |
| 8 | **Salivary Imunoglobulina A**<br>in 6% (6/94) of Studies and 20% (1/5) of Parameters with Veracity Metrics | Concentration of IgA | (n = 1) | CV = 3.7–91.0% | (Small–Large) |
| 9 | **Blood Lactate Concentration**<br>in 5% (5/94) of Studies and 0% (0/8) of Parameters with Veracity Metrics | | | | |
| 10 | **Video-Computerized System**<br>in 5% (5/94) of Studies and 0% (0/11) of Parameters with Veracity Metrics | | | | |
| 11 | **Watch/Wristwatch**<br>in 5% (5/94) of Studies and 50% (1/2) of Parameters with Veracity Metrics | Trained/played minutes (min) | (n = 1) | CV = 5.5% | (Small) |
| 12 | **Body Composition**<br>in 3% (3/94) of Studies and 0% (0/3) of Parameters with Veracity Metrics | | | | |
| 13 | **Integrative Tool of Training Load Assessment**<br>in 3% (3/94) of Studies and 0% (0/3) of Parameters with Veracity Metrics | | | | |
| 14 | **Repeated Sprint Ability**<br>in 3% (3/94) of studies and 38% (3/8) of parameters with veracity metrics | Mean (s)<br>Best (s)<br>Decrement (%) | (n = 1)<br>(n = 1)<br>(n = 1) | ICC = 0.61<br>ICC = 0.69<br>ICC = 0.34 | (Moderate)<br>(Moderate)<br>(Poor) |
| 15 | **Salivary Cortisol**<br>in 3% (3/94) of Studies and 50% (1/2) of Parameters with Veracity Metrics | Concentration of cortisol | (n = 1) | CV = 3.1% | (Small) |
| 16 | **Salivary Testosterone**<br>in 3% (3/94) of Studies and 100% (1/1) of Parameters with Veracity Metrics | Concentration of testosterone | (n = 1) | CV = 4.1% | (Small) |
| 17 | **Submaximal Running Test**<br>in 3% (3/94) of Studies and 0% (0/2) of Parameters with Veracity Metrics | | | | |
| 18 | **Sprint Test**<br>in 3% (3/94) of Studies and 0% (0/6) of Parameters with Veracity Metrics | | | | |
| 19 | **Anaerobic Speed Reserve**<br>in 2% (2/94) of Studies and 36% (4/11) of Parameters with Veracity Metrics | Meters covered at maximal aerobic speed (m)<br>Time spent at maximal aerobic speed (min)<br>Meters covered at 30% anaerobic speed reserve (m)<br>Time spent at 30% anaerobic speed reserve (min) | (n = 1)<br>(n = 1)<br>(n = 1)<br>(n = 1) | MDC = 116–144%<br>MDC = 112–174%<br>MDC = 73–145%<br>MDC = 7–116% | |
| 20 | **CK**<br>in 2% (2/94) of Studies and 33% (1/3) of Parameters with Veracity Metrics | CK absolute concentration (μ/L) | (n = 1) | SEM = 2.3% | (Good) |
| 21 | **Maximal Oxygen Uptake**<br>in 2% (2/94) of Studies and 0% (0/4) of Parameters with Veracity Metrics | | | | |
| 22 | **Visual Analogue Scale Questionnaire**<br>in 2% (2/94) of Studies and 0% (0/7) of Parameters with Veracity Metrics | | | | |
| 23 | **Total Quality Recovery**<br>in 2% (2/94) of Studies and 50% (1/2) of Parameters with Veracity Metrics | Recovery level (Morandi et al., 2020) | (n = 1) | SEM = 30.0%<br>CV = 12.0%<br>ICC = 0.77 | (Poor)<br>(Medium)<br>(Good) |

**Table 1.** *Cont.*

| | | | | | |
|---|---|---|---|---|---|
| 24 | **Training Planning** in 2% (2/94) of Studies and 0% (0/7) of Parameters with Veracity Metrics | | | | |
| 25 | **Actigraphy** in 1% (1/94) of Studies and 13% (1/8) of Parameters with Veracity Metrics | Total sleep time | (n = 1) | CV = 10.0% | (Small) |
| 26 | **Blood Ammonia Concentration** in 1% (1/94) of Studies and 0% (0/1) of Parameters with Veracity Metrics | | | | |
| 27 | **Coach rating of Performance** in 1% (1/94) of Studies and 0% (0/1) of Parameters with Veracity Metrics | | | | |
| 28 | **Edinburgh Mental Well-being Scale** in 1% (1/94) of Studies and 0% (0/1) of Parameters with Veracity Metrics | | | | |
| 29 | **Infrared Thermography** in 1% (1/94) of Studies and 0% (0/1) of Parameters with Veracity Metrics | | | | |
| 30 | **Isometric Force testing** in 1% (1/94) of Studies and 100% (2/2) of Parameters with Veracity Metrics | Total peak force relative to body weight (N/kg) at 30° and 90° | (n = 1) | CV = 7.5% | (Small) |
| 31 | **Neuromuscular Efficiency Index** in 1% (1/94) of Studies and 0% (0/3) of Parameters with Veracity Metrics | | | | |
| 32 | **Oxidative Stress** in 1% (1/94) of Studies and 0% (0/1) of Parameters with Veracity Metrics | | | | |
| 33 | **Physical Activity Enjoyment Scale** in 1% (1/94) of Studies and 0% (0/1) of Parameters with Veracity Metrics | | | | |
| 34 | **Profile of Mood State Questionnaire (POMS)** in 1% (1/94) of Studies and 100% (1/1) of Parameters with Veracity Metrics | Total mood disturbance | (n = 1) | ICC = 0.21 | (Poor) |
| 35 | **RESTQ-Sport Scale** in 1% (1/94) of Studies and 0% (0/4) of Parameters with Veracity Metrics | | | | |
| 36 | **Sit-and-Reach Test** in 1% (1/94) of Studies and 0% (0/1) of Parameters with Veracity Metrics | | | | |
| 37 | **SJ Test** in 1% (1/94) of Studies and 100% (1/1) of Parameters with Veracity Metrics | Jump height (cm) | (n = 1) | ICC = 0.95 | (Excellent) |
| 38 | **Tensiomyography** in 1% (1/94) of Studies and 100% (4/4) of Parameters with Veracity Metrics | Maximum radial muscle belly displacement Contraction time Delay time Half-relaxation time | (n = 1) (n = 1) (n = 1) (n = 1) | ICC = 0.97 ICC = 0.98 ICC = 0.87 ICC = 0.80 | (Excellent) (Excellent) (Good) (Good) |
| 39 | **Urine Metabolomic** in 1% (1/94) of Studies and 100% (17/17) of Parameters with Veracity Metrics | (a) steroid hormone metabolites: hydrocortisol, tetrahydrodeoxycortisol, dihydrotestosterone glucuronide, androsterone glucuronide, cortolone-3-glucuronide, testosterone glucuronide, tetrahydroaldosterone-3-glucuronide; (b) hypoxanthines: hypoxanthine, 8-hydroxy-7-methylguanine; (c) acetylated amino acids: N-acetylglutamic acid, phenylalanyl-aspartic acid; (d) intermediates in phenylalanine metabolism: 2-phenylacetamide, phenylacetic acid; (e) tyrosine and indolic tryptophan metabolites: indole-3-carboxylic acid, indolepyruvic acid; (f) riboflavin: vitamin B2 and 4-pyridoxic acid. | (n = 1) | Precision = 3–10% | |

## 4. Discussion

The purpose of this systematic review is to evaluate data veracity for load monitoring in professional soccer. We describe here the metrics reported for tools and their associated parameters in a sample with 87% of athletes playing at top-level divisions or in their national teams. Thirty-nine different tools were found in the included studies; however, 73% of these studies did not report any veracity metric. Thus, data veracity was found for 54% of tools and 23% of associated parameters. Of note, some veracity metrics present

a great variability in the obtained metrics. The SEM and MDC ratings were between 0.3–89.7% (good–poor) and 48.0–305.0%, respectively. The ICC ratings were between poor and excellent reliability levels (i.e., 0.004–0.98), with the CV ratings from poor to great (i.e., 1.1–193.5%). For the remaining metrics, the results were between 2.0–10.0% for bias, 4.0–22.0% for SEE, and 3.0–10.0% for precision. In addition, some tools were used without the required accuracy. All this information provides a precise state of the art regarding the potential of current monitoring tools to be used with Big Data approaches.

### 4.1. The Impact of Accuracy and CV on the Practical Application of Tools and Parameters

The need for reporting accuracy can be better understood following an example reported by the United Nations Industrial Development Organization [24]. Thus, in practical terms, if the accuracy of a tool is ±3, this means that if 100 would be the reading displayed on a tool during a measurement, then the actual value could be anywhere between 97 and 103, including 97 and 103. In this regard, it is important to note that some tools did not present the required accuracy for the experimental design performed in each study. For example, the accuracy of an infrared camera is ±2 °C, and the applied protocol in the study provided for changes in body temperature ranged between 0.3 and 1.5 °C [98]. Moreover, in EPTS with a sampling rate of 10 Hz, the accuracy for high accelerations (>4 m/s$^2$) is compromised, which may have an impact on the interpretation of results [79] because of the poor estimation of instantaneous velocity when performing these very high accelerations (>4 m/s$^2$) during player tracking in team sports [135]. Therefore, tools with inadequate accuracy for the required interventions need to improve their accuracy or else they willgenerate an input noise, thus compromising the outputs of a Big Data approach.

A common concern of sports scientists as well as strength and conditioning coaches of high-level soccer clubs when using monitoring tools refers to reliability, which is one of the main factors related to the discrepancy between the expected and actual effectiveness of monitoring, thus representing a potential barrier for successful interventions [136]. For this purpose, the classic classification of the CV is considered for the decision-making process: >30% = large, <30% and >10% = medium, and <10% = small [40]. In the present systematic review, extremely large CVs, such as 193.5% for sprinting at speeds ≥25.2 km/h, are measured with EPTS [129]. Moreover, 137% for the time between peak power and peak force during a CMJ [137] and 679% for the variance from mean bedtime assessed by Actigraphy [138] have been found in the scientific literature. Large CVs make it extremely difficult to detect the real differences between moments after an intervention unless these differences were also very large [139]. Therefore, caution should be taken when using parameters with a CV greater than 30%.

### 4.2. ICC and SEM Should Always Be Reported Together

The ICC is probably the most popular reliability metric in the literature. Our analysis showed that the ICC was often included as a reliability metric but with large variability, ranging from poor = 0.004 (i.e., strain from session-RPE) [131] to excellent = 0.98 (i.e., contraction time measured with tensiomyography or jump height of a CMJ) [57,131]. Particularly, in the case of tensiomyography, although there is sufficient data in favor of its good-to-excellent relative reliability in the scientific literature (i.e., ICC = 0.70–0.99) [140] in agreement with our findings (i.e., ICC = 0.80–0.98), more evidence is necessary for identifying the accuracy of this tool, according to a current systematic review with meta-analysis [141], mainly because reliability based only on the ICC cannot be recommended [142]. This is because there is a relationship between the ICC and between-subjects variability, where the heterogeneity of the subjects is a determinant for the ICC obtained [25]. An excellent ICC can mask poor trial-to-trial consistency when between-subjects variability is high. Conversely, a poor ICC can be found even when trial-to-trial variability is low when the between-subjects variability is low. In this case, the homogeneity of the subjects' means will make it difficult to differentiate between subjects even though the absolute measurement error is small. In other words, if individuals differ little from each other, the

ICC may be poor, even if the trial-to-trial variability is small [25]. Thus, an examination of the ICC in conjunction with the SEM is, therefore, recommended [142]. For example, anaverage HR, with ICC = 0.77 (good) and SEM = 3.0% (good), in one case [127] would be preferable than the same parameter with ICC = 0.89 (good) and SEM = 27% (poor) [124].

On the other hand, the SEM is not affected by between-subjects variability, as occurs with the ICC. The SEM has been used to define the boundaries in which a subject's true score lies, and it can be calculated as both absolute and relative scores [25,26,37,38]. However, the range of variation of this veracity metric for the found parameters was high (i.e., between good = 0.3% for total distance covered, as measured with EPTS, and poor = 89.7% for accelerations >4 m/s$^2$, measured with EPTS) [64,73]. This would suggest that for the same tool, some parameters could be more recommended than others. In this regard, some researchers are using a limit of 10% of the SEM to determine which parameters can be used in subsequent analyses [35,87,143]. Although its use is strongly recommended by the specialized literature [25,26], some points of confusion can be highlighted. The first is that it is called either standard error of measurement (SEM), typical error of measurement (TEM), or typical error (TE), even with the same form of calculation. The second point, which is most impactful, is that the SEM is commonly referred to as the coefficient of variation (CV) and reported as a percentage, but the calculation is not performed by dividing the standard deviation by the mean and subsequently multiplied by 100 [144]. These facts can easily drive researchers and practitioners to misinterpretations. Furthermore, the SEM can also be used in the interpretation of individual scores as the minimum difference (MD) or minimum detectable change (MDC) [25,26]. In the present study, the MDC ranged from 48.0% for the distance at maximal sprint speed recorded with EPTS to 305.0% for the very high-speed running time obtained with EPTS [76] with regards to the group average. This extremely high variability could hinder the verification of real changes in an individual's performance. Of note, the MDC in professional soccer players is not easily found in scientific literature. Apart from the included study in the current systematic review, another study reported an MDC between 1.0% and 30.0% for performance in a new agility test [145]. Therefore, the SEM is highly recommended for determining the reliability of any tool and parameter, while its interpretation can be facilitated when reported in both relative and absolute terms.

*4.3. Needs, Limitations, and Potential of Big Data in Professional Soccer*

There is limited information regarding the last three metrics of veracity. Bias was reported as absolute values in only two studies [127,129], and, in one study, it was reported as relative values ranging between 2.0–10.0%, with the SEE ranging between 4.0–22.0% for total and sprint distances between 19.8–25.2 and above 25.2 km/h, recorded with EPTS [133]. Precision ranged between 3.0–10.0% for urine metabolomic parameters in a single study [128]. In view of the definition itself, these three metrics are important for appropriately understanding veracity. Therefore, more studies with the use of these metrics in professional soccer athletes are warranted as well as more studies with professional female soccer players, who represented only 5% of the samples in the included studies. Moreover, 94% of the included articles are observational studies, and among the experimental study designs, none were randomized controlled trials. Despite the difficulty of carrying out experimental studies in professional soccer settings, we suggest that more of these studies are warranted to increase the veracity of monitoring tools and their associated parameters. The approach applied in this systematic review is based on one of the 4 Vs of Big Data. However, the other Vs also have an impact on soccer and should be the focus of future research. In this regard, the volume that describes the magnitude of the data [14] and is usually measured in terabytes (i.e., $10^{12}$), petabytes (i.e., $10^{15}$), zettabytes (i.e., $10^{21}$), and even yottabytes (i.e., $10^{24}$) [15] has increased exponentially, with a 300-fold increase in the last 15 years, from 2005 to 2020, reaching 40 zettabytes (i.e., 37.3 trillion gigabytes; $10^9$) [15]. Therefore, this unprecedented volume of data is overwhelming, thus increasing the risks of it not being fully used to inform the practice [13]. For example, a

dataset from a Bundesliga season resulted in 400 gigabytes of tracking data [14]. Regarding this, variety refers to the heterogeneity of data, i.e., to different data formats and sources distinguished, which can be differentiated among structured (e.g., relational data), semi-structured, (e.g., XML data), and unstructured data (e.g., emails, pictures, videos, or social networking data) [14–17]. Thus, in soccer, data variety refers to position, video, fitness, training, skill performance, and health data [14]. Moreover, velocity describes the speed at which novel data is generated, processed, and analyzed [14–17]. Specifically for soccer, the velocity may vary between real-time streams from physiological and positional data to stored data tonotational analyses during training and competition [14]. All these three key concepts characterizing Big Data are highly relevant and should be considered with the suggestion by Lukoianova and Rubin [23], who have stated that Big Data can only have value when its veracity can be established and, thereby, the information quality confirmed.

To date, Big Data has been reported in the scientific literature of soccer in relation to tactical analysis only [14,146] but not in relation to load monitoring. Although data can be considered big based on the number of V2019s, there are no universal benchmarks for the number of Vs [14–18,21–23,146,147]. Therefore, the current systematic review establishes the initial basis for the implementation of Big Data approaches for load monitoring in professional soccer while identifying the strengths and limitations of the current evidence before moving on to the real applications of Big Data: data management, which involves processes and supporting technologies to acquire and store data and to prepare and retrieve it for analysis, and analytics, which refers to the techniques used to analyze and acquire intelligence from Big Data [146,147]).

Finally, we would like to make a brief statement and endorse an editorial on the current trend of having extreme positions in sportsscience, which is not recommended [148]. Thus, this review highlights the importance of data veracity analysis for objectively using tools and their associated parameters for load monitoring in professional soccer, considering equally traditional approaches and the forefront of technological advances.

## 5. Conclusions

In conclusion, a wide diversity of tools and parameters are used to monitor loads in elite professional soccer. However, it is not common to find data veracity for these tools and parameters in scientific literature. The reported veracity metrics will assist in the selection and use of the best monitoring tools and their associated parameters for load monitoring in professional soccer. Before looking for new tools and parameters, the current ones need to present adequate levels of data veracity (accuracy, reliability, and quality of data). Therefore, this information is warranted when aiming to use predictive analytics for structured Big Data in professional soccer.

*Practical Applications*

1. The use of Big Data approaches without appropriate data veracity can undermine the precision of the predictive analytics models and generate fatal errors with a high economic cost;
2. Up to the moment, data veracity is not commonly reported in scientific studies using tools and parameters for load monitoring in professional soccer;
3. It is necessary to more frequently analyze and share data veracity to perform the best data management and analytics when applying Big Data.

**Supplementary Materials:** The following are available online at https://www.mdpi.com/article/10.3390/app11146479/s1, Table S1: Risk of bias score; Table S2: Summary of the selected studies. Reference [149] is cited in supplementary materials.

**Author Contributions:** Conceptualization, J.G.C., C.A.C.F., D.B., and J.C.S.; methodology, J.G.C., C.A.C.F., D.B., A.L.-A., G.R.C., R.L.d.S.G., R.d.S.G., F.M.V., A.L.C.A., S.D.R., and J.C.S.; formal analysis, J.G.C., C.A.C.F., D.B., A.L.-A., G.R.C., R.L.d.S.G., R.d.S.G., F.M.V., A.L.C.A., S.D.R., J.A., and J.C.S.; writing—original draft preparation, J.G.C., C.A.C.F., D.B., A.L.-A., G.R.C., R.L.d.S.G., R.d.S.G., F.M.V., A.L.C.A., S.D.R., J.A., and J.C.S.; writing—review and editing, J.G.C., C.A.C.F., D.B., A.L.-A., S.D.R., J.A., and J.C.S.; supervision, J.C.S. All authors have read and agreed to the published version of the manuscript.

**Funding:** This research received no external funding.

**Institutional Review Board Statement:** Not applicable.

**Informed Consent Statement:** Not applicable.

**Data Availability Statement:** After publication, all data necessary to understand and assess the conclusions of the manuscript are available to any reader of Applied Sciences.

**Acknowledgments:** We would like to thank the "Coordenação de Aperfeiçoamento de Pessoal de Nível Superior/ Programa de ExcelênciaAcadêmica" (CAPES/PROEX).

**Conflicts of Interest:** The authors declare no conflict of interest.

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
