# Peer review of "The Role of Veracity on the Load Monitoring of Professional Soccer Players: A Systematic Review in the Face of the Big Data Era"

_applsci, doi:10.3390/app11146479_

Round 1

Reviewer 1 Report

The authors wrote an interesting manuscript where they analyzed data veracity for the load monitoring in professional soccer players. The expertise of the authors is evident from the text.

The content of the “Introduction” is relevant to the topic and generally provides a quality background for the research.

Part “Materials and Methods” is comprehensible, and the methods used for systematic review are adequate.

“Results” are presented in a well-arranged way. The “Discussion” and “Conclusions” are conducted within the framework of the obtained results.

However, results include a few discrepancies which should be explained.

Line 199: “averacity” should be “a veracity”

Line 205: Most tools for load monitoring are straightforward. There is a tool named “Watch”. Could the authors be more specific about this tool? Did the authors mean the method of video analysis or observation? Please explain.

Line 222: Table S2 is in the supplementary document – this information should be clearly stated in the text; the table is difficult to locate.

In conclusion, the manuscript is well-written. Except for a few shortcomings, this is a high-quality systematic review with practical applicability.

Author Response

Journal: Applied Sciences

Manuscript Status: Pending minor revisions

Manuscript ID: applsci-1251991

Type: Systematic Review

Dear Editor Elaina Li, 

We would like to thank you and the reviewers for their kind comments on our paper. We have addressed the points raised and we would like to re-submit the revised version of the manuscript "The role of veracity on load monitoring of professional soccer players: a systematic review in the face of the Big Data Era". An item-by-item response is presented below. All changes in the manuscript are highlighted in red color. We appreciate the contributions from these set of reviews and we believe these amendments have improved the quality of our paper.

Sincerely yours,

João Gustavo Claudino and co-Authors

REVIEWER #1:

Comments and Suggestions for Authors:

The authors wrote an interesting manuscript where they analyzed data veracity for the load monitoring in professional soccer players. The expertise of the authors is evident from the text.

The content of the “Introduction” is relevant to the topic and generally provides a quality background for the research.

Part “Materials and Methods” is comprehensible, and the methods used for systematic review are adequate.

“Results” are presented in a well-arranged way. The “Discussion” and “Conclusions” are conducted within the framework of the obtained results.

We would like to thank this Reviewer for her/his kind comments. We have amended the manuscript accordingly.

However, results include a few discrepancies which should be explained.

Line 199: “averacity” should be “a veracity”

Changed accordingly and one space was inserted.

Line 205: Most tools for load monitoring are straightforward. There is a tool named “Watch”. Could the authors be more specific about this tool? Did the authors mean the method of video analysis or observation? Please explain.

Changed accordingly and the word “Wristwatch” was added across the manuscript.

Line 222: Table S2 is in the supplementary document – this information should be clearly stated in the text; the table is difficult to locate.

Changed accordingly and this sentence was added in the main manuscript: “Specifically Table S2 presents a summary of all selected studies with further information on study design (and duration), sample level (n; sex; age; country of sample), tools (brand and model or reference), accuracy reported by the company, parameters as well as veracity analysis (metrics).”

In conclusion, the manuscript is well-written. Except for a few shortcomings, this is a high-quality systematic review with practical applicability.

Thank you so much again for your kind comments and suggestions.

Reviewer 2 Report

No coments

Author Response

Journal: Applied Sciences

Manuscript Status: Pending minor revisions

Manuscript ID: applsci-1251991

Type: Systematic Review

Dear Editor Elaina Li, 

We would like to thank you and the reviewers for their kind comments on our paper. We have addressed the points raised and we would like to re-submit the revised version of the manuscript "The role of veracity on load monitoring of professional soccer players: a systematic review in the face of the Big Data Era". An item-by-item response is presented below. All changes in the manuscript are highlighted in red color. We appreciate the contributions from these set of reviews and we believe these amendments have improved the quality of our paper.

Sincerely yours,

João Gustavo Claudino and co-Authors

REVIEWER #2:

Comments and Suggestions for Authors:

No coments

We would like to thank this Reviewer for her/his kind comments.

Reviewer 3 Report

I have reviewed the article titled "The role of veracity on load monitoring of professional soccer 2 players: a systematic review in the face of the Big Data Era" This is a very interesting article. Presentation of the findings should be improved.   

Though systematically reviewed, it is important that some aspects of the findings be summarized and systematically presented.  It will be important that key insights be summarized in ways that the reader can visualize where possible. It is also important that key areas be summarized in text sections.

I understand that the PRISMA guidelines (Moher et al. 2009) were followed for deciding which elements should be included in the systematic review. However, these guidelines are a bit outdated considering recent developments in analysis capabilities. 

For presenting the results of this systematic study, it would be useful if the authors of the manuscript could add 'italicized' subsections outlining important areas more systematically.  Additionally it would be helpful if some findings of the authors could be summarized in 'leading tables'. 

Some examples are shown in the following three more recent articles. These articles also refer to other recent methods-related papers for bibliometric analysis within and in the reference section.

Authors of the manuscript can see section 4.2 in the recent article by GA Duffy (cited below) and section 4.3 in the recent article by BM Duffy for specific examples of 'italicized subsections'.  Examples of leading tables for their respective topics areas are shown throughout both articles. 

Authors of the MDPI manuscript can see section 4.5 in the following more recent article for specific examples of 'italicized subsections'.

Kurniawan, Jessica, and VG Duffy. "Systematic Review of the Importance of Human Factors in Incorporating Healthcare Automation." In International Conference on Human-Computer Interaction, pp. 96-110. Springer, Cham, 2021. Available at SpringerLink.

Author Response

Journal: Applied Sciences

Manuscript Status: Pending minor revisions

Manuscript ID: applsci-1251991

Type: Systematic Review

Dear Editor Elaina Li, 

We would like to thank you and the reviewers for their kind comments on our paper. We have addressed the points raised and we would like to re-submit the revised version of the manuscript "The role of veracity on load monitoring of professional soccer players: a systematic review in the face of the Big Data Era". An item-by-item response is presented below. All changes in the manuscript are highlighted in red color. We appreciate the contributions from these set of reviews and we believe these amendments have improved the quality of our paper.

Sincerely yours,

João Gustavo Claudino and co-Authors

REVIEWER #3

Comments and Suggestions for Authors

I have reviewed the article titled "The role of veracity on load monitoring of professional soccer 2 players: a systematic review in the face of the Big Data Era" This is a very interesting article. Presentation of the findings should be improved.   

Though systematically reviewed, it is important that some aspects of the findings be summarized and systematically presented.  It will be important that key insights be summarized in ways that the reader can visualize where possible. It is also important that key areas be summarized in text sections.

We would like to thank this Reviewer for her/his kind comments. We have amended the manuscript accordingly.

I understand that the PRISMA guidelines (Moher et al. 2009) were followed for deciding which elements should be included in the systematic review. However, these guidelines are a bit outdated considering recent developments in analysis capabilities. 

Changed accordingly: the PRISMA was updated.

For presenting the results of this systematic study, it would be useful if the authors of the manuscript could add 'italicized' subsections outlining important areas more systematically.  Additionally it would be helpful if some findings of the authors could be summarized in 'leading tables'. 

Some examples are shown in the following three more recent articles. These articles also refer to other recent methods-related papers for bibliometric analysis within and in the reference section.

Authors of the manuscript can see section 4.2 in the recent article by GA Duffy (cited below) and section 4.3 in the recent article by BM Duffy for specific examples of 'italicized subsections'.  Examples of leading tables for their respective topics areas are shown throughout both articles. 

Authors of the MDPI manuscript can see section 4.5 in the following more recent article for specific examples of 'italicized subsections'.

Kurniawan, Jessica, and VG Duffy. "Systematic Review of the Importance of Human Factors in Incorporating Healthcare Automation." In International Conference on Human-Computer Interaction, pp. 96-110. Springer, Cham, 2021. Available at SpringerLink.

Changed accordingly: the manuscript was updated based on the Duffy’s paper (Guo F, Liu L, Lv W, Li F, Duffy VG. A bibliometric analysis of occupational low back pain studies from 2000 to 2020. Arch Environ Occup Health. 2021 Mar 3:1-10. doi: 10.1080/19338244.2021.1893634. Epub ahead of print. PMID: 33653232). The access to the book chapter was not allowed for us.